# Different Effects of 12-Week Speed or Accuracy Training on Obstacle-Crossing Foot Motion in Healthy Elderly

**DOI:** 10.3390/ijerph19084596

**Published:** 2022-04-11

**Authors:** Yusuke Maeda, Daisuke Sudo, Daiki Shimotori

**Affiliations:** 1Department of Physical Therapy, School of Health Sciences at Odawara, International University of Health and Welfare, Kanagawa 250-8588, Japan; d.sudo@iuhw.ac.jp; 2Komaki City Hospital, Aichi 485-8520, Japan; shimo.daiki.040127@gmail.com

**Keywords:** obstacle crossing, motion speed, accuracy of motion

## Abstract

Preventing falls is important in the elderly. One reason for falling is tripping or stumbling; hence, it is important to improve the crossing motion. This study aimed to compare speed- and accuracy-oriented crossing training and establish a useful training method. To investigate the effects of crossing motion training, we conducted a randomized controlled trial. Twenty healthy elderly individuals (aged 71.7 ± 1.5 years) were randomly assigned to two groups: speed training and accuracy training groups. They practiced initiating their crossing motion faster or more accurately for 12 weeks. Using a three-dimensional motion analysis system, the data on the crossing motion was captured before and after the training period. We set four conditions (normal speed, fast, leaning stance, and leaning stance and fast) and two directions (anterior and lateral) to analyze the crossing motion. The crossing motion of the speed training group became significantly faster compared to baseline (*p* < 0.05); however, the accuracy of the crossing motion of the accuracy training group was not statistically significant. Speed training in this study had clear effects on crossing motion. It is surprising that crossing motion training from a normal upright stance can also improve swing speed from the leaning stance. We believe that this training is easy and useful in the elderly population.

## 1. Introduction

Falls are one of the most serious problems faced by the elderly population. According to previous studies, tripping and stumbling are common reasons for such accidents [1,2,3]. Proper obstacle-crossing motion requires the integration of sensory inputs and outputs of the joint torque for sufficient foot clearance. Due to muscle weakness [4,5], sensory dysfunction [6,7], and cognitive impairment [8], proper obstacle avoidance motion deteriorates in the elderly.

To avoid tripping or stumbling and to prevent falls, it is important to step swiftly and accurately. If elderly people can swing their legs fast, they are more likely to avoid falling. Previous studies revealed that elderly population with fall histories showed slowed volitional steps [9,10]. The reaction time for the crossing motion of elderly people was shown to be longer than that of youngsters [11]. Additionally, the elderly population takes longer durations to perceive postural changes because of an increase in somatosensory thresholds. To compensate for the delay in perception, it is important for the elderly people to achieve a rapid crossing motion.

Similarly, an accurate crossing is also important. Elderly people lack “motion consistency.” The definition of accurate crossing motion in this study is the ability of people to swing their foot as intended. If this condition is met, then the person’s foot motion should be consistently the same no matter how many times the crossing motion is attempted. Moreover, we define motion consistency as “how well the foot trajectories match” when the crossing motion is performed multiple times. In this study, we evaluated motion consistency by overlaying foot trajectories for 10 times of crossing; therefore, the exact definition of motion consistency is the degree of similarity of foot trajectories when an individual performs the crossing motion 10 times. If the foot trajectories of 10 crossing motions mostly match, the motion should be judged as “intended.” However, if they do not match, the motion should be judged as “unintended” and as having “a lack of motion consistency.” For example, if an elderly person attempts the same motion multiple times and every time their motion is different, we consider this motion as a “inconsistent.” Elderly people find it difficult to place their foot through the same root of space, or to land their foot on the same place. Given the above, if the elderly take a step by unintended motion when they are about to fall, it is difficult for them to make a new base of support. Therefore, we recognize motion inconsistency as a high risk of fall. We have previously demonstrated that the trajectories of the heel during the crossing motion in elderly people were inconsistent [12]. Some studies have reported that the foot clearance in the elderly population during crossing is larger than that in young adults [13,14]. However, another study reported a smaller value [15]. Therefore, the motion consistency of the elderly population was not stable.

It is important to consider the proper crossing motion not only from the upright standing position, but also from the tilting stance position. We believe that it is difficult for the elderly people to step when the body is tilted or leaning. A previous study reported that elderly people cannot recover with a single step swiftly from the forward leaning position, compared with young adults [16]. Robinovitch et al. reported that it is difficult for fallers to shift their weight properly [17]. Therefore, it is important to investigate the crossing motion of the elderly individuals recovering from the leaning position to the upright position.

To improve their crossing motion, it is necessary to train the elderly on a daily basis. We believe that volitional crossing is the best way to perform home training because it is easy, simple, and convenient. The purpose of this study was to compare the effects of speed- and accuracy-crossing training at home using a randomized controlled trial and to establish a practical training method. We hypothesized that both training methods would affect the crossing motion and be useful in home-based training.

## 2. Materials and Methods

### 2.1. Participants

Twenty healthy elderly individuals (6 men and 14 women, aged 71.7 ± 1.5 years, height 153.5 ± 4.2 cm, and weight 51.6 ± 6.8 kg) without neurological and musculoskeletal diseases were enrolled in this study. The participants were informed with a written explanation of the experimental purpose, procedures, potential risks, and the right to refuse inclusion in the study. All participants provided written informed consent to participate in this study. The experimental procedures were approved by the ethical committee of the International University of Health and Welfare (17-Io-43).

We recruited participants through the “Public interest incorporated association Odawara silver center.” All participants were independent and were working once or twice a week. The exclusion criteria of this study were: history of stroke; severe diabetes; dementia; and need of walking aids for locomotion. Once the exclusion criteria had been considered, the registered members were contacted by the center by telephone. They were randomly assigned to two groups of 10 people each (speed training and accuracy training groups; STG and ATG, respectively) (Figure 1). Table 1 indicates the participants’ characteristics. Each individual participated in the experiment before the intervention. After the 12-week intervention, the same experiment was performed.

### 2.2. Intervention

Participants were instructed to perform obstacle-crossing motion 50 times in the anterior and lateral directions per day. They did this intervention 3 days a week for 12 weeks. They did not have to decide the exact day of the week and could perform the motion as appropriate, depending on their physical condition, for a total of 12 weeks. To prevent falls, as a precaution, they were instructed to perform the training at the corner of the room, and to not perform it in the leaning position. The STG was also asked to perform the motion as fast as possible. The ATG was asked to perform the same motion as much as possible. This means that the foot went through the same trajectory and landed in the same position 50 times. The participants could rest at any time, depending on when they felt fatigued. The exact motion that had to be performed was the same as the one in the experiment conducted before the training; therefore, we could make sure that the participants could understand the motion during the experiment. To ensure the prosecution of the task, we recorded the date of the training day.

### 2.3. Procedure

We have conducted the same procedure in a previous study [12]. We instructed the participants to cross over the obstacle, which was 15% of their height. The obstacle was a light wooden stick (about 60 × 2 × 2 cm), supported by an iron pipe. The dominant foot was used for the crossing. The dominant foot was defined as the side of kicking a ball. The right foot was dominant in all the participants. The obstacle was placed 10 cm forward and lateral to the participants’ feet (Figure 2A,B). They crossed the obstacle with their right leg from upright standing position and then stopped the motion after landing. They were instructed not to look at their feet during the motion. We set four conditions and two directions to analyze the crossing motion.

Normal condition: crossing over the obstacle from upright stance position with normal, natural speed;Fast condition: crossing over the obstacle from the upright stance position as fast as possible;Leaning condition: crossing over the obstacle from leaning posture with normal, natural speed;Leaning-fast condition: crossing over the obstacle from leaning posture as fast as possible.

The participants crossed the obstacle 10 times in each condition. There were two crossing directions (anterior and lateral), and each participant performed a total of 80 trials (10 trials × 4 conditions × 2 directions). To set the leaning position, we measured the limit of stability using the force plate of each participant, in advance. They attempted to move their body forward and right from an upright standing posture to their maximum possible extended position without moving their feet. The maximum leaning position of the center of foot pressure (COP) was defined as 100% in two directions. In the leaning and leaning-fast conditions, the participants waited at a 70% leaning position to the anterior or lateral, and then crossed over the obstacle with the examiner’s cue. The examiner monitored the COP position in real time to maintain the participants in the 70% leaning position for 1–2 s, and then gave the cue for crossing motion (Figure 3A). If the participants tripped or stumbled on the obstacle, the trial was excluded. The participants practiced the motion for each condition several times.

The data on the crossing motion was captured using a three-dimensional analysis system (Vicon Motion System Ltd., Yarnton, UK). The Nexus 2.0 software was used for recording and monitoring. Nine cameras and four force plates were synchronized. The sampling rates of the cameras and force plate were 100 Hz and 1000 Hz, respectively. We defined Y as the anterior direction (anterior, +), X as the lateral direction (right, +), and Z as the vertical direction (upward, +). Reflective markers (14 mm in diameter) were placed on the right heel, lateral malleolus, and the head of second metatarsal bone. We calculated the center position of the swing foot (G foot) in the horizontal plane using the X and Y coordinates of the reflective markers (Figure 3B).
$$G\;\mathrm{foot}=(\frac{X1+X2+X3}{3},\;\frac{Y1+Y2+Y3}{3});\;$$

The X and Y coordinates were below;

Right heel (X1, Y1, Z1), lateral malleolus (X2, Y2, Z2), the head of second metatarsal bone (X3, Y3, Z3).

### 2.4. Data Analysis

The motion analysis software Visual 3D (C-motion, Inc., Germantown, MD, USA) was used to analyze the motion capture data. The data on the coordinates of the reflective markers during the swing phase were extracted. The start and end points of the data were defined as the moment when the heel marker moved 2 mm (heel off) and the force plate detected 10 N of foot contact (landing). Each data point was normalized from 0 to 1. To calculate the average trajectory of the 10 data points, the curves were approximated using a cubic spline function, and the data were resampled at regular intervals. Figure 4 shows the calculation method or sum of the distance error of the heel. The d(τ) was calculated as indicated below:$$d\left(\tau \right)=\left|f\left(\tau \right)-\bar{f\left(\tau \right)}\right|$$

d(τ) represents the gap between each trial and average at time τ. In addition, $\bar{d\left(\tau \right)}$ is the average of d(τ) across 10 trials, normalized from 0 to 1. The average of d(τ) was averaged by 10. In addition, crossing speed was calculated as the time from heel off to landing. To investigate the accuracy of the landing position, the X and Y coordinates of the swing foot at landing were recorded.

d(τ) is defined as indicated below:$$d\left(\tau \right)=\left|f\left(\tau \right)-\bar{f\left(\tau \right)}\right|$$

d(τ) represents the gap between each trial and average at time τ. In addition, $\bar{d\left(\tau \right)}$ is the average of d(τ) across 10 trials, normalized from 0 to 1.

The differences in each parameter of the sum of the distance error and crossing speed were compared before and after the training period using paired *t*-test. Statistical significance was defined as *p* < 0.05. The *p* value, effect size (*r*), and 95% confidence intervals (CIs) were calculated. Effect size (*r*) was calculated as shown below:$$r=\sqrt{\frac{t^{2}}{t^{2}+df}}$$

*t*: *t*-value, *df*: degrees of freedom

All statistical analyses were conducted using SPSS 25.0 (IBM, Inc., Endicott, NY, USA).

## 3. Results

All participants completed the 12-week intervention. Table 2 and Table 3 indicate the average distance error (m) of the anterior crossing of the STG and ATG groups, respectively. In the leaning condition, the STG significantly improved the accuracy of the heel trajectory (*p* < 0.01); however, the ATG showed no statistical differences after the intervention. Table 4 and Table 5 indicate the average distance error of the lateral crossing of the STG and ATG groups, respectively. Both training groups showed no statistically significant differences after the intervention.

Table 6 and Table 7 indicate the crossing time (s) of anterior crossing in the STG and ATG groups, respectively. The crossing time of the STG in the anterior direction was significantly shortened by 13.1%, 18.4%, 13.5%, and 15.4%, in the normal, fast, leaning, and leaning and fast conditions, respectively (*p* < 0.05). Similarly, in the normal condition, the crossing time in the ATG was shortened by 7% (*p* < 0.05). Table 8 and Table 9 indicate the lateral crossing speed of the STG and ATG groups, respectively. The crossing time of the STG in the lateral direction was significantly shortened by 10.3%, 11.5%, 11.6%, and 8.2%, in the normal, fast, leaning, and leaning and fast conditions, respectively (*p* < 0.05). Notably, the effect size of the leaning and fast conditions of the STG was the largest (Table 8). In contrast, the ATG showed no statistical differences in the crossing speed for lateral crossing.

Table 10 and Table 11 indicate the standard deviation (SD) of 10 times the X coordinate of the landing position of the STG and ATG groups for anterior crossing, respectively. A large number of SDs indicates a large variation of 10 times the landing. There were no significant differences in any of the conditions for anterior crossing. Table 12 and Table 13 indicate the SD of 10 times of Y coordinate of the landing position of the STG and ATG groups for anterior crossing, respectively. While the STG indicated a significant difference in the leaning and fast conditions for lateral crossing (*p* < 0.05), there were no significant differences in the other conditions. Table 10, Table 11, Table 12 and Table 13 can be found below:

## 4. Discussion

This study demonstrated that step time was shortened about 11–19% for anterior direction and 8–11% for lateral direction as a result of the home training. These results indicated that speed training is effective for crossing motions. The speedup of the motion indicates that each joint moment of the leg also increases because the acceleration and deceleration of the joint require a force. Lamoureux et al. reported a significant association between the lower muscle strength and the obstructed gait ability [18]. Fast crossing motion requires a larger joint moment of the lower leg to lift the lower extremities, compared to normal crossing or common gait. We believe that both the speedup of the crossing motion and strengthening of the lower leg muscles occurred because of the repeated speed training. Lord et al. reported that fall accidents were related to a lack of fast voluntary steps [10]. Therefore, speed training in this study may have decreased the risk of falls.

The crossing speed in the leaning position was also improved by the crossing training from the upright standing position. It is difficult to cross over rapidly from the leaning stance because body weight is loaded in the direction to step, especially in the lateral direction. In this experiment, the participants mostly loaded on the right foot for lateral crossing immediately before the crossing motion. They returned their weight to the center once, and then lifted their right leg. This weight shift is challenging for elderly individuals. In terms of lateral stepping, Mille et al. reported that elderly individuals tend to take a cross-over step by an unloaded leg when they are disturbed in the lateral direction [19]. The authors considered that the fast lateral step by the loaded leg required a large torque from the hip abductor muscles [19]. Therefore, hip abductors may have been strengthened by speed training in this study.

A previous study reported that balance training, including crossing motion, significantly reduced COP sway and improved timed up and go test score and fast gait speed [20]. We believe that single-leg standing can be improved by crossing motion training. The stance leg must support the entire body during the swing of the opposite leg. In this case, a larger inertial force was generated during unstable standing by fast swing. Advanced postural control ability is required to maintain a one-leg stance while taking a larger inertial force. Therefore, postural control ability can be improved with speed training. According to a previous study, fallers require more steps to recover postural stability against disturbances than young adults [21]. In this study, we made the participants stop their motion immediately after crossing. When the participants try to cross quickly, the inertial force increases. It is difficult for them to stop immediately after landing; therefore, postural control ability could be improved by this speed training. Moreover, the speedup of the motion at the leaning position has a great advantage in avoiding fall accidents. Elderly individuals tend to take longer to perceive a leaning posture because of a decline in sensory function. A previous study suggested that there was a significant association between prolonged latency of the step and fall rates [22]. To compensate for the delay in sensory perception, both fast swing and weight shift are required. In the forward leaning position, body weight is loaded on the forefoot. To cross fast, it is necessary to shift the body weight to the support leg and then lift the crossing leg quickly. It is difficult for the elderly to shift quickly and appropriately. High-speed crossing can build a new base of support quickly when the center of gravity is going outside of the current limit of stability. This training method has been shown to be easy and effective. In-home training should be safe, considering the absence of attendants during training. The crossing motion in the leaning position was less stable than that in upright standing. It is useful for in-home elderly population that crossing training from an upright stance, not a leaning stance, could improve crossing motion.

However, accuracy training has no significant effect on crossing motion in both anterior and lateral directions. A significant decrease in the distance error was not observed with accuracy training. Moreover, improved consistency of the landing position has not been found. According to previous research, it is difficult for elderly individuals to modify their targeted foot landing position, which moves unexpectedly [23]. In this experiment, the participants had no target for landing position, and we instructed them to look forward during the crossing motion. A previous study suggested that a proper trajectory of the limb for a step requires visual information [24]. The authors also reported that visual information was used for feed-forward planning of limb movements [25]. We believe that the landing position was inconsistent, even after accuracy training, because of the lack of visual information.

This study has few limitations. We did not collect data on muscle strength of the lower extremities, foot length, and postural control ability. The reasons for the increase in crossing speed are unclear. In future, data on muscle strength or one-leg stance ability should be collected. These data provide valid evidence for the effects of crossing motion training.

## 5. Conclusions

This study aimed to compare the effects of speed and accuracy training in the elderly population. The STG and ATG groups performed their designated training for 3 days a week and 12 weeks, respectively. While accuracy training had limited effect after the training period, speed training had clear effects on the crossing speed, with step time in the STG being significantly shortened by about 13–18% and 8–11% for the anterior and lateral directions, respectively. The practice of crossing from a normal upright stance resulted in a speedup of the crossing motion from the leaning stance. These results indicate that this method focused on the motion speed is easy and useful for in-home elderly individuals. Further studies are required in order to investigate why speed training has a positive effect on the crossing motion in the elderly, and how the crossing motion’s accuracy can be improved in this setting.

## Figures and Tables

**Figure 1 ijerph-19-04596-f001:**
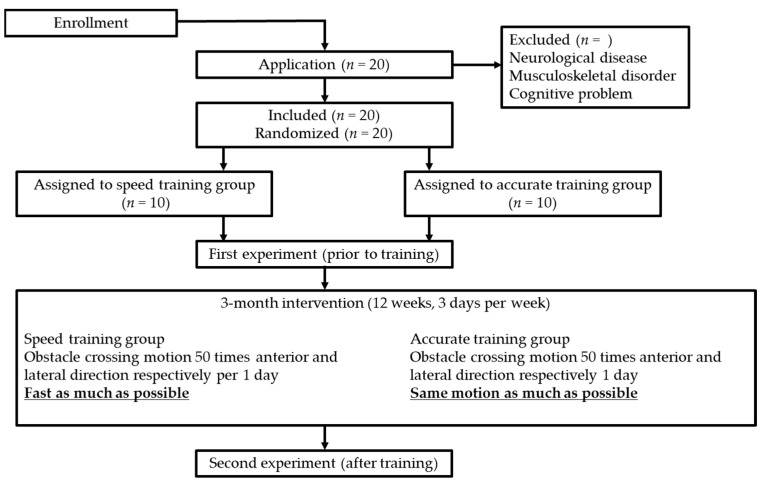
Participant flowchart from enrollment to intervention and experiment.

**Figure 2 ijerph-19-04596-f002:**
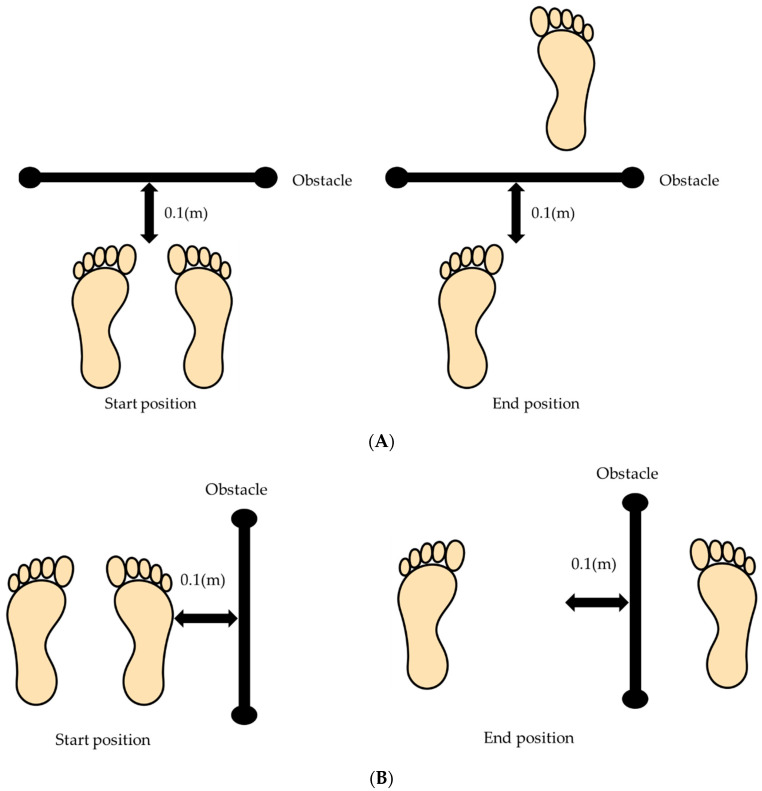
(**A**). The start and end foot position in the anterior crossing movement. (**B**). The start and end foot position in the lateral crossing movement.

**Figure 3 ijerph-19-04596-f003:**
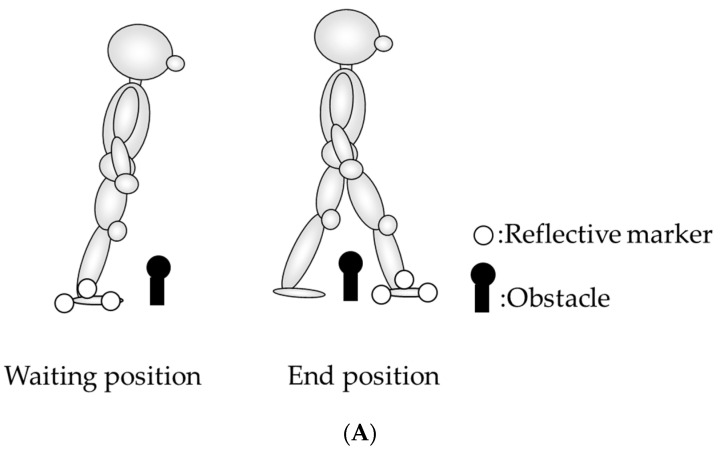
(**A**). The participants’ waiting and end positions. (**B**). Reflective marker coordinates of the swing foot (G foot).

**Figure 4 ijerph-19-04596-f004:**
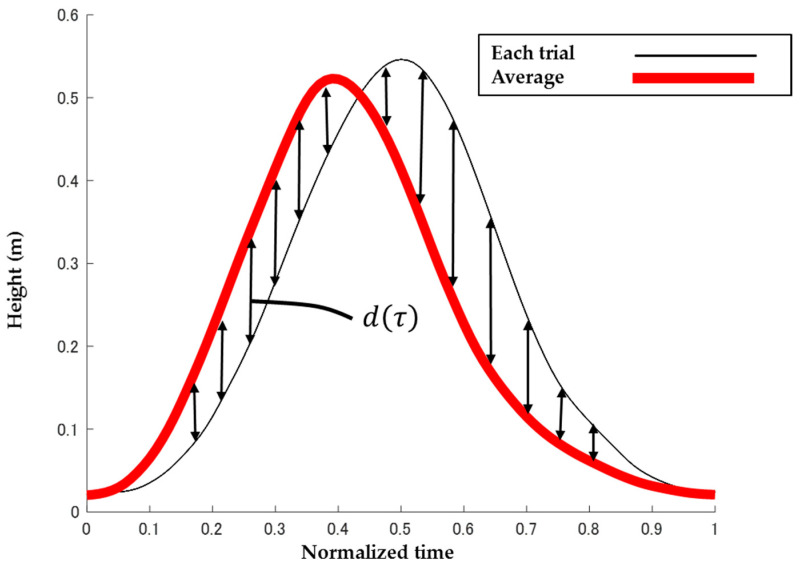
Example of calculation of the distance error.

**Table 1 ijerph-19-04596-t001:** Participants’ characteristics.

	STG (*n* = 10)	ATG (*n* = 10)
Age (years)	71.4 ± 1.3	71.9 ± 1.7
Height (cm)	153.1 ± 5.1	153.9 ± 3.5
Weight (kg)	51.7 ± 7.1	51.5 ± 6.8
Body mass index	22.0 ± 2.4	21.8 ± 3.2
Number of right-footed	10	10
Number of men	3	3
Mean obstacle height (cm)	23.0 ± 0.6	

**Table 2 ijerph-19-04596-t002:** Average distance error in the STG for heel anterior obstacle crossing.

		Averaged Distance Error (m)	95%CI	Effect Size (*r*)	*p* Value
Normal	Before	0.0155 ± 0.0044	−0.026–0.05	0.23	0.49
	After	0.0143 ± 0.0031			
Fast	Before	0.0147 ± 0.0064	−0.029–0.083	0.35	0.29
	After	0.0120 ± 0.0040			
Leaning	Before	0.0171 ± 0.0048	−0.025–0.071	0.34	0.30
	After	0.0148 ± 0.0048			
Leaning and Fast	Before	0.0140 ± 0.0035	0.016–0.07	0.77	<0.01
	After	0.0098 ± 0.0017			

**Table 3 ijerph-19-04596-t003:** Average distance error in the ATG for heel in anterior crossing.

		Averaged Distance Error (m)	95%CI	Effect Size (*r*)	*p* Value
Normal	Before	0.0129 ± 0.0024	−0.018–0.038	0.25	0.45
	After	0.0119 ± 0.0021			
Fast	Before	0.0111 ± 0.0041	−0.016–0.045	0.33	0.31
	After	0.0097 ± 0.0023			
Leaning	Before	0.0162 ± 0.0065	−0.010–0.079	0.51	0.11
	After	0.0127 ± 0.0041			
Leaning and Fast	Before	0.0110 ± 0.0029	−0.017–0.023	0.11	0.75
	After	0.107 ± 0.032			

**Table 4 ijerph-19-04596-t004:** Average distance error in the STG for heel in lateral obstacle crossing.

		Averaged Distance Error (m)	95%CI	Effect Size (*r*)	*p* Value
Normal	Before	0.0164 ± 0.0029	−0.019–0.033	0.20	0.55
	After	0.0156 ± 0.0060			
Fast	Before	0.0166 ± 0.0073	−0.287–0.087	0.36	0.28
	After	0.0137 ± 0.0053			
Leaning	Before	0.0213 ± 0.0089	−0.040–0.115	0.34	0.30
	After	0.0176 ± 0.0060			
Leaning and Fast	Before	0.0177 ± 0.0101	−0.065–0.124	0.23	0.49
	After	0.0148 ± 0.0067			

**Table 5 ijerph-19-04596-t005:** Average distance error in the ATG for heel in lateral obstacle crossing.

		Averaged Distance Error (m)	95%CI	Effect Size (*r*)	*p* Value
Normal	Before	0.0206 ± 0.0107	−0.009–0.138	0.55	0.08
	After	0.0142 ± 0.0305			
Fast	Before	0.0145 ± 0.0059	−0.129–0.082	0.17	0.62
	After	0.0169 ± 0.0141			
Leaning	Before	0.0201 ± 0.0065	−0.040–0.080	0.24	0.47
	After	0.0181 ± 0.0049			
Leaning and Fast	Before	0.0148 ± 0.0050	−0.022–0.035	0.17	0.61
	After	0.0141 ± 0.0059			

**Table 6 ijerph-19-04596-t006:** Crossing time in the STG for heel in anterior obstacle crossing.

		Crossing Time (s)	95%CI	Effect Size (*r*)	*p* Value
Normal	Before	0.927 ± 0.059	0.037–0.235	0.72	<0.01
	After	0.806 ± 0.110			
Fast	Before	0.764 ± 0.112	0.054–0.210	0.79	<0.01
	After	0.624 ± 0.046			
Leaning	Before	0.920 ± 0.115	0.040–0.214	0.74	<0.01
	After	0.796 ± 0.037			
Leaning and Fast	Before	0.735 ± 0.079	0.047–0.165	0.81	<0.01
	After	0.622 ± 0.052			

**Table 7 ijerph-19-04596-t007:** Crossing time in the ATG for heel in anterior obstacle crossing.

		Crossing Time (s)	95%CI	Effect Size (*r*)	*p* Value
Normal	Before	0.878 ± 0.081	0.007–0.124	0.65	<0.05
	After	0.821 ± 0.120			
Fast	Before	0.635 ± 0.075	−0.043–0.055	0.09	0.80
	After	0.633 ± 0.091			
Leaning	Before	0.790 ± 0.074	−0.012–0.084	0.49	0.13
	After	0.757 ± 0.089			
Leaning and Fast	Before	0.626 ± 0.077	−0.010–0.068	0.48	0.14
	After	0.601 ± 0.048			

**Table 8 ijerph-19-04596-t008:** Crossing time in the STG for heel in lateral obstacle crossing.

		Crossing Time (s)	95%CI	Effect Size (*r*)	*p* Value
Normal	Before	0.855 ± 0.083	0.026–0.132	0.75	<0.01
	After	0.776 ± 0.100			
Fast	Before	0.674 ± 0.092	0.039–0.101	0.86	<0.01
	After	0.603 ± 0.063			
Leaning	Before	0.805 ± 0.116	0.001–0.185	0.61	<0.05
	After	0.712 ± 0.079			
Leaning and Fast	Before	0.636 ± 0.105	0.0001–0.085	0.61	<0.05
	After	0.584 ± 0.065			

**Table 9 ijerph-19-04596-t009:** Crossing time in the ATG for heel in lateral obstacle crossing.

		Crossing Time (s)	95%CI	Effect Size (*r*)	*p* Value
Normal	Before	0.802 ± 0.100	−0.056–0.068	0.07	0.84
	After	0.797 ± 0.093			
Fast	Before	0.619 ± 0.063	−0.011–0.054	0.45	0.17
	After	0.618 ± 0.046			
Leaning	Before	0.760 ± 0.087	−0.059–0.053	0.04	0.90
	After	0.763 ± 0.101			
Leaning and Fast	Before	0.624 ± 0.069	−0.021–0.051	0.29	0.38
	After	0.609 ± 0.048			

**Table 10 ijerph-19-04596-t010:** Movement variation in the STG in anterior (SD, Y) and lateral (SD, X) direction of G-foot anterior obstacle crossing.

		(SD, X) (m)				(SD, Y) (m)			
			95%CI	Effect Size (*r*)	*p* Value		95%CI	Effect Size (*r*)	*p* Value
Normal	Before	0.019 ± 0.009	−0.019–0.009	0.45	0.17	0.408 ± 1.225	−0.480–1.260	0.32	0.34
	After	0.015 ± 0.005				0.019 ± 0.012			
Fast	Before	0.014 ± 0.004	−0.011–0.001	0.49	0.12	0.022 ± 0.012	−0.002–0.014	0.13	0.64
	After	0.019 ± 0.009				0.026 ± 0.024			
Leaning	Before	0.017 ± 0.006	−0.006–0.007	0.04	0.90	0.021 ± 0.008	−0.001–0.014	0.55	0.08
	After	0.016 ± 0.007				0.015 ± 0.005			
Leaning and Fast	Before	0.015 ± 0.006	−0.001–0.005	0.45	0.16	0.018 ± 0.009	−0.001–0.011	0.50	0.12
	After	0.013 ± 0.004				0.013 ± 0.003			

**Table 11 ijerph-19-04596-t011:** Movement variation in the ATG in anterior (SD, Y) and lateral (SD, X) direction of G-foot anterior obstacle crossing.

		(SD, X) (m)				(SD, Y) (m)			
			95%CI	Effect Size (*r*)	*p* Value		95%CI	Effect Size (*r*)	*p* Value
Normal	Before	0.011 ± 0.007	−0.007–0.002	0.32	0.34	0.019 ± 0.009	−0.005–0.010	0.23	0.49
	After	0.013 ± 0.005				0.017 ± 0.005			
Fast	Before	0.014 ± 0.005	−0.001–0.004	0.36	0.27	0.015 ± 0.004	−0.022–0.010	0.27	0.41
	After	0.012 ± 0.006				0.021 ± 0.023			
Leaning	Before	0.013 ± 0.004	−0.024–0.009	0.32	0.33	0.016 ± 0.007	−0.023–0.011	0.26	0.44
	After	0.021 ± 0.024				0.022 ± 0.024			
Leaning and Fast	Before	0.012 ± 0.004	−0.004–0.003	0.13	0.68	0.013 ± 0.002	−0.002–0.003	0.14	0.69
	After	0.013 ± 0.004				0.013 ± 0.003			

**Table 12 ijerph-19-04596-t012:** Movement variation in the STG in anterior (SD, Y) and lateral (SD, X) direction of G-foot lateral obstacle crossing.

		(SD, X) (m)				(SD, Y) (m)			
			95%CI	Effect Size (*r*)	*p* Value	Landing Position SD (X)	95%CI	Effect Size (*r*)	*p* Value
Normal	Before	0.017 ± 0.004	−0.004–0.007	0.18	0.59	0.015 ± 0.007	−0.006–0.006	0.01	0.99
	After	0.016 ± 0.005				0.015 ± 0.006			
Fast	Before	0.018 ± 0.006	−0.005–0.007	0.12	0.72	0.016 ± 0.006	−0.002–0.009	0.42	0.19
	After	0.017 ± 0.008				0.013 ± 0.005			
Leaning	Before	0.036 ± 0.038	−0.004–0.047	0.53	0.09	0.024 ± 0.018	−0.003–0.023	0.49	0.12
	After	0.015 ± 0.005				0.014 ± 0.006			
Leaning and Fast	Before	0.018 ± 0.005	0.001–0.009	0.67	0.03	0.017 ± 0.004	−0.004–0.009	0.20	0.42
	After	0.013 ± 0.005				0.014 ± 0.006			

**Table 13 ijerph-19-04596-t013:** Movement variation in the ATG in anterior (SD,Y) and lateral (SD, X) direction of G-foot lateral obstacle crossing.

		(SD, X) (m)				(SD, Y) (m)			
			95%CI	Effect Size (*r*)	*p* Value	Landing Position SD (X)	95%CI	Effect Size (*r*)	*p* Value
Normal	Before	0.036 ± 0.033	−0.003–0.045	0.54	0.09	0.030 ± 0.025	−0.001–0.035	0.57	0.07
	After	0.015 ± 0.006				0.013 ± 0.004			
Fast	Before	0.029 ± 0.044	−0.014–0.047	0.37	0.26	0.025 ± 0.028	−0.007–0.033	0.42	0.18
	After	0.013 ± 0.003				0.012 ± 0.004			
Leaning	Before	0.033 ± 0.025	−0.001–0.035	0.59	0.06	0.023 ± 0.018	−0.001–0.023	0.55	0.08
	After	0.015 ± 0.005				0.013 ± 0.005			
Leaning and Fast	Before	0.023 ± 0.021	−0.008–0.025	0.35	0.29	0.021 ± 0.017	−0.004–0.020	0.44	0.17
	After	0.014 ± 0.005				0.013 ± 0.003			

95%CI: 95% confidence intervals. *r* = $\sqrt{\frac{t^{2}}{t^{2}+df}}$.

## Data Availability

The datasets analyzed during this study are available from the corresponding author upon reasonable request.

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
