# Peer review of "Different Effects of 12-Week Speed or Accuracy Training on Obstacle-Crossing Foot Motion in Healthy Elderly"

_ijerph, 2022, doi:10.3390/ijerph19084596_

Round 1

Reviewer 1 Report

Journal                      International Journal of Environmental Research and Public Health

Effect of speed and accuracy training in obstacle-crossing in the elderly

General comments

For this original article authors analyzed a great volume of data with practical outcomes in training of elderly people with the aim to prevent falls.  

The main goal of the study was to compare the 12-week effect between speed- and accuracy training on crossing motion of dominant foot in four different conditions (normal and fast speed, leaning stance and leaning stance with fast speed) using 3D instrumented gait analysis. Participants were 20 healthy persons with mean age 71.7 year randomly distributed in two groups with different training performing stepping exercises with obstacles (15% of participant’s height) 50 times once per day three times a week.

Description of aim, methods and results is clear enough, but needs specification.  The references give an overview of the studies in the topic of research. Data is presented in four figures and 12 tables.

Specific comments

Title of the manuscript: it is recommended to  change the title to reflect the main idea and subjects. For example, “Different effects of 12-month speed- or accuracy training in obstacle-crossing on foot motion in healthy elderly”.

Please use this in abstract and description of main findings and in conclusions.

Abstract

L12 Please add the age of participants

Introduction

L60-61. Merge both hypotheses in one sentence

Materials and Methods

Participants

Please add data of two groups in Table; gender (and number, % of men in each group), age (mean, range, SD), body height, body mass, body mass index, and if you have, length of foot,

dominant foot.

It is recommended to use abbreviation of name of groups in all text- SPEED-TRAINING GROUP (STG) AND ACCURACY TRAINING GROUP (ATG) and use the same formulation in all text.

Please add information about physical activity of persons if you have – did they participate in fitness or any sport activity (their physical load – times per day, per week), was their work connected to physical load or did they not work anymore.

Please add exclusion criteria here and number of excluded subjects, also in Fig 1.

Fig 1 should be here.

Have the participants had any previous injuries, othopaedical or neurological diseases/operation of lower extremity, cardiovascular or pulmonological diseases, peripheral artery disease, diabetes, vision impairments/using correction glasses, foot pathology/operations?

Please check the ethics committee permission document – 17-Io-43, are there numbers or letters in the middle, what is the year of the permission.

Please add the full name of recruitment center and ways of recruitment – phone calls, e-mails, letters etc.

Was the experiment done at morning/evening time? How long did the pre-post study take? Was it done at the same time of day?

Were participants familiarized with the study and exercises performance?

Intervention

Please give more information about intervention procedures.

L76-77. Please correct the sentence – what are these 57 directions?

L79 as much as possible or as exactly/accurately as possible? Please comment in text.

Fig 2 should be here (Fig 2A and 2B instead of 2-1 and 2-2).

Where was the intervention performed? Did participants exercise at home for 12 weeks – if so, how did they check their leaning position?  Please add the exact description of the leaning position – how many degrees it was by mean?

Did they keep a diary? How was the performance feedback organized for the participants? Was the same physiotherapy specialist involved or how was the training session organized? How long was it in minutes?

Please add a full description of obstacle crossing exercises here -  by numbers of exercises and starting position.

How was the weight distribution on two legs at the end of crossing?

Did you use any verbal motivation/correction of position?

Procedures

Please distribute this part into three:

2.3.1. Preparation for obstacle crossing exercise

Please provide mean height of obstacles and SD  in Table 1.

Please add also the length and width of obstacles

L84 What about non-dominant foot position and its position at the start and finish of one exercise?

Same questions: How was the weight distribution on two legs at the end of crossing?

Did you use any verbal motivation/correction of position?

2.3.1 Leaning position setting

L 108 … did the examiner keep a cue with inclination during the start of stepping? – please add a figure here of participant position and foot with markers (Fig 3A).

Did the examiner test the 70% leaning of the body by COP each day before training/ before each exercise?

Which device and software were used to study COP sway?

2.3.3. Foot motion measurement

Why four force plates – did participants step over any obstacles also – how many obstacles and how long walkway were in this case?

Nine infrared cameras?

Fig 3 should be here, please use the title: “Reflective markers coordinates of the swing foot (G foot)” (Fig 3B).

L115, 120 please use term “the head of second metatarsal bone”

The mean data of 10 trials was taken for analysis? Please add comment

Data analysis

Fig 4 formula should be here in the text.

L189  May be better: “…across ten trials, where obstacle crossing step time cycle normalized from 0 to 1”?.

Fig 4 should be here, for example: “Example of calculation of the distance error (explanation in text)”.

Please use words “distance error” instead of error distance

Results

L139. All participants were able to perform the test at normal and fast speed? No falls/discomfort? Add comments please.

L148  By how much % did the speed increase? Please use % to indicate data dynamics.

Tables 1,2. Please merge Tables 1 and 2, and correct the title as follows: Table 1. Average distance error for G-foot anterior obstacle crossing

  1. Speed training group
  2. Accuracy training group

Also it is better to use data in cm, not in m.

Table 3,4.  Please merge Tables 3 and 4, and correct the title as follows: Table 2. Average distance error for G-foot lateral obstacle crossing

  1. Speed training group
  2. Accuracy training group

Also it is better to use data in cm, not in m.

Tables 5,6. Please merge Tables 5 and 6, and correct the title as follows: Table 3. Time of  G-foot anterior obstacle crossing

  1. Speed training group
  2. Accuracy training group

Tables 7,8. Please merge Tables 7 and 8, and correct the title as follows: Table 4. Time of  G-foot lateral obstacle crossing

  1. Speed training group
  2. Accuracy training group

Tables 9,10. Please merge Tables 9 and 10, and correct the title as follows: Table 5. Movement variation in anterior (SD,Y) and lateral (SD, X) direction of G-foot anterior obstacle crossing

  1. Speed training group
  2. Accuracy training group

Please add unit – cm (better)

Tables 11,12. Please merge Tables 11 and 12, and correct the title as follows: Table 6. Movement variation in anterior (SD,Y) and lateral (SD, X) direction of G-foot lateral obstacle crossing

  1. Speed training group
  2. Accuracy training group

Please add unit – cm (better)

L242-243 – formula ???? If you need, please insert the formula in the text of Statistics.

Discussion.

Please begin the chapter with main results of the present study – see my comment above; as well as paragraphs.

L289 – add full name of abbr. COG

Please use same terms – see above: SPEED-TRAINING GROUP (STG) AND ACCURACY TRAINING GROUP (ATG); SPEED-TRAINING AND ACCURACY TRAINING

L306 Limitations – foot length? Control of COM?

Please add here comments about two hypotheses.

Conclusions:

Please comment on your main results here only and questions for future studies.

Author's contribution : DS – participated only in two activities for manuscript preparation?

References:

  1. Please use abbr. of journal name
  2. Who are the authors?

Author Response

Dear Reviewer,

We would like to thank you for reviewing in detail our manuscript (manuscript ID 1610616) entitled “Different effects of 12-week speed- or accuracy-training on obstacle-crossing foot motion in healthy elderly”.

We hereby submit a revised manuscript conforming to all of the Reviewers’ comments and suggestions. In particular, we have addressed all the Reviewers’ comments in a point-by-point manner, and the revisions performed are underlined in the revised manuscript. Moreover, please note that we have modified the title of our manuscript, according to Reviewer 1’s suggestion.

Please find enclosed the revised manuscript. We hope that the revised manuscript is now suitable for publication in International Journal of Environmental Research and Public Health.

Thank you for your consideration. We look forward to hearing from you.

Sincerely,

Yusuke Maeda, PhD

Reviewer 2 Report

This study is about the effect of speed and accuracy training in obstacle-crossing in the elderly.

I have some questions as below.

  1. In the introduction, a few concepts have been mentioned, such as the proper obstacle crossing motion, accurate crossing, motion consistency. They should be explained in more details so we can have better understanding of the dependent variables of this study.
  2. Page 2, Line 72: Why was there no control group in this study?
  3. Page 2, Line 78-79: In the intervention, the speed-training group was also asked to perform the motion as fast as possible. The accurate-training group was asked to perform the same motion as much as possible. How did they make sure all the participants strictly followed the instructions? Also, the requirement was unclear and very subjective. Why was it expected to be effective for obstacle crossing? Any references?
  4. Page 2, Line 83-86: No references have been provided for the procedure. Have any previous studies conducted similar measures before? Why this procedure?
  5. Page 3, Line 134: Only the differences before and after the intervention have been tested using paired t-test. Are there any differences between speed-training group and accurate-training group? Why?
  6. Figure 2-1: The figure can not be seen in this format.
  7. Tables: Too many tables have been presented in this paper. Some of them can be merged or put in the supplemental materials.

Author Response

(The authors gave the same response as above.)

Reviewer 3 Report

Thank you for the opportunity to read this manuscript. This is a very interesting study.

However, the authors should rephrase or explain on detail.

How did the authors instruct (give verbal cues) for the training to the participants? It is an important information to use. The statistical analyses should be reconsidered.

Line 18: faster than?

Line 33: The sentence should be reconsidered because other factors also must relate to prevention of falling.

Line 45: The sentence is not clear to understand and needs to be rephrased.

Line 63: Was the sample size calculated? Please provide it.

Line 78: As for the speed training, the task may lose balance resulting to fall.

Line 92: Fast condition and speed training must be the same. Please unify either one in the entire manuscript.

Line 98: How about the effects of fatigue?

Figure 2-1: I could not see the figure 2-1. Please check and correct.

Figure 2-2: The figure is not easy to understand the foot movement. Please reconsider and correct.

Line 134: The statistics need to be reconsidered. Two-way ANOVA should be conducted instead of t-tests.

Line 264: Put reference of Mille et al.

Line 265: cross-over step

Line 290: The authors stated the home training is safe, but the number of repetition and the speed training leads to fatigue and/or balance perturbation.

Line 312: The sentence of Twenty healthy… is not needed in the conclusion. Please erase.

Line 317: Please be more specific of “this method.”

Author Response

(The authors gave the same response as above.)

Round 2

Reviewer 2 Report

All questions have been addressed in the paper.